# Degree of adaptive response in urban tolerant birds shows influence of habitat-of-origin

Lawrence E. Conole

School of Geography and Environmental Studies, University of Tasmania, Sandy Bay, Tasmania, Australia

## ABSTRACT

Urban exploiters and adapters are often coalesced under a term of convenience as 'urban tolerant'. This useful but simplistic characterisation masks a more nuanced interplay between and within assemblages of birds that are more or less well adapted to a range of urban habitats. I test the hypotheses that objectively-defined urban exploiter and suburban adapter assemblages within the broad urban tolerant grouping in Melbourne vary in their responses within the larger group to predictor variables, and that the most explanatory predictor variables vary between the two assemblages. A paired, partitioned analysis of exploiter and adapter preferences for points along the urban–rural gradient was undertaken to decompose the overall trend into diagnosable parts for each assemblage. In a similar way to that in which time since establishment has been found to be related to high urban densities of some bird species and biogeographic origin predictive of urban adaptation extent, habitat origins of members of bird assemblages influence the degree to which they become urban tolerant. Bird species that objectively classify as urban tolerant will further classify as either exploiters or adapters according to the degree of openness of their habitats-of-origin.

## INTRODUCTION

The community of ecologists studying urban bird ecology has to a large extent converged on *Blair*'s (*1996*) typology of 'urban exploiters', 'suburban adapters' and 'urban avoiders', defined by the bird assemblages' biological and behavioural traits (*Chace & Walsh, 2006*; *González-Oreja et al., 2007*; *Kark et al., 2007*; *Croci, Butet & Clergeau, 2008*). Such assemblages as described here are elsewhere sometimes characterised as 'response guilds' (*Leveau, 2013*). Exploiters and adapters are often coalesced under a term of convenience as 'urban tolerant'. Such a useful but simplistic characterisation of the urban tolerant subset may mask a more nuanced interplay between and within groups of birds that are more or less well adapted to a range of urban habitats, ranging from the intensely urbanised 'down town' areas of the inner city, out through a fluctuating gradient of generally decreasing urbanisation intensity through the suburbs to the urban fringe. That there are identifiable

Corresponding author
Lawrence E. Conole,
lconole@gmail.com

'exploiters' and 'adapters' in addition to the 'avoiders' suggests further targeted testing of the urban tolerant grouping may be fruitful in understanding some underlying processes in urban bird ecology.

A humped distribution of bird species richness has been observed in a number of urban studies, with highest values recorded in the intermediate urbanisation intensity range on the rural–urban gradient (*Tratalos et al., 2007*; *Luck & Smallbone, 2010*; *Shanahan et al., 2014*). This pattern has been shown to hold true for all species, but also for urban tolerant species as a subset (*Shanahan et al., 2014*). However, results of earlier data analyses of Melbourne birds suggest that the two assemblages within the urban tolerant group may not show the uniform response to urbanisation as has been shown for other cities (*Conole, 2011*; *Conole & Kirkpatrick, 2011*).

Gradient analysis (*Ruszczyk et al., 1987*) has been broadly applied in urban ecological studies over the past two decades (*McDonnell & Hahs, 2008*), and much longer in ecology more generally (*Whittaker, 1967*). It is intuitively compatible with a landscape ecology perspective (*Snep, Timmermans & Kwak, 2009*), and despite criticisms of the limitations of gradient analysis as an approach for studying urban ecology (*Catterall, 2009*; *Ramalho & Hobbs, 2012a*), the potential remains for this approach to be the 'scaffolding' upon which deeper investigations are built (*McDonnell, Hahs & Pickett, 2012*; *Ramalho & Hobbs, 2012b*). In taking the assemblages identified through gradient analysis (*Conole & Kirkpatrick, 2011*) as the basis for the present study, I acknowledge the reality that the urban–rural gradient is not simplistically linear (*Ramalho & Hobbs, 2012a*) or neatly concentric around the 'down town' centre (*Catterall, 2009*). The reality of non-concentricity does not limit the usefulness of gradient analysis in understanding complexity and nuance in urban bird ecology. While acknowledging the utility of the urban exploiter/adapter typology, I seek in this paper to deconstruct the concept of 'urban tolerance' for birds, and test the hypothesis which contends that 'urban tolerance' is not monolithic, but multifaceted.

The urban tolerance status of birds included in many published studies has been applied *a priori*, based on work of others in geographically related systems (such as *Kark et al., 2007*), or compiled from secondary or tertiary descriptive sources (such as *Bonier, Martin & Wingfield, 2007*, but see *González-Oreja et al., 2007*). It is also the case that many urban bird studies are largely descriptive or narrowly site-specific (*Marzluff, Bowman & Donnelly, 2001*; *McDonnell & Hahs, 2013*), lacking either a theoretical underpinning or focus (*Scheiner, 2013*), and there have been calls to formulate research questions designed to develop a greater mechanistic understanding of the underlying ecological processes operating in urban landscapes (*Shochat et al., 2006*; *McDonnell & Hahs, 2013*), and move towards generalisable concepts (*Mac Nally, 2000*).

Part of the process of moving towards generalisable concepts in urban bird ecology involves gaining a better understanding of the extent to which the degree of adaptation to urban environments progresses from intolerance to the high level of adaptation that characterises exploiters. How similar are the responses of the adapters and exploiters to different aspects of the urban–rural gradient?

The data in this paper are focused on two assemblages characterised by the author as urban exploiters and suburban adapters from Melbourne, Australia (Fig. S1) (*Conole & Kirkpatrick, 2011*). The present study departs from the approach taken in many others of similar kind in that urban bird assemblages that form the basis of the work were objectively classified at the landscape scale from direct data analyses (*Conole & Kirkpatrick, 2011*) rather than indirect inference or *a priori* assignment. I attempt a paired, partitioned analysis of exploiter and adapter preferences for points along the urban–rural gradient to decompose the overall trend into diagnosable parts for each assemblage, in a way not previously seen in the literature.

I test the hypotheses that the distinct urban exploiter and suburban adapter assemblages within the broad urban tolerant grouping in Melbourne vary in their responses to predictor variables. I also test the hypothesis that habitat-of-origin has predictive utility in determining which urban tolerant birds become exploiters or adapters.

## MATERIALS & METHODS

Detailed descriptions of the study area and methodology used to derive the urban bird assemblages can be found in *Conole & Kirkpatrick (2011)*, and are summarised in the Supplemental Text.

### Study area and data handling

The study area is metropolitan Melbourne; capital city of the State of Victoria in coastal southeastern Australia, within a 50 km radius of its Central Business District (Fig. S1) (37°49′S and 144°58′E).

Approximately 220,000 records of birds were extracted from the Birds Australia 'New Atlas of Australian Birds' database (*Barrett et al., 2003*), and intersected with a $1 \times 1$ km grid (*Hahs & McDonnell, 2006*) to produce a matrix of grid cells by species presence/absence. Species and sites were filtered out according to criteria for representativeness (see Supplemental Text) to arrive at a final list of 141 species and 390 cells (*Conole & Kirkpatrick, 2011*).

### Environmental and demographic indices

Spatial data on the degree of urbanisation of the study area employed in this study were developed at ARCUE and are discussed in detail by *Hahs & McDonnell (2006)*; a brief summary of the two selected factors follows.

Frequency Greenspace (hereafter greenspace) is the reciprocal of the average amount of impervious surface calculated at the sub-pixel level from the impervious surface fraction image created during the spectral mixture analysis of the 2000 Landsat ETM + image (*Hahs & McDonnell, 2006*).

Combined index ($Index_{Combined}$) is the average value of $Index_{Image}$ and $Index_{Census}$; where $Index_{Image}$ is calculated from fraction images produced by the spectral mixture analysis of the 2000 Landsat ETM + image, and $Index_{Census}$ = the total number of people multiplied by the proportion of males employed in non-agricultural work, as enumerated in the 2001 Australian census (*Hahs & McDonnell, 2006*).

Other environmental factors considered in analyses included PC_URB (percent cover of urban landform), People per square kilometre (People/km$^2$—the total number of people in census collection districts) and Dwellings per square kilometre (Dwellings/km$^2$—the total number of houses in census collection districts) (*Hahs & McDonnell, 2006*; *Conole & Kirkpatrick, 2011*).

## Data analysis

Statistical analyses were performed in R (*R Core Team, 2013*) using base R functions and procedures from the R-packages 'vegan' (*Oksanen et al., 2013*) and 'bayespref' (*Fordyce et al., 2011*). Figures were drawn using R base graphics, R-packages 'vegan' and 'ggplot2' (*Wickham, 2009*; *Oksanen et al., 2013*), and QGIS (*QGIS Development Team, 2013*).

An earlier assemblage analysis (*Conole & Kirkpatrick, 2011*) was the basis for partitioning the total bird datasets for this study; detailed methodology is described therein. Adapter and exploiter species were further partitioned into two new matrices for this study, and separate non-metric multidimensional scaling (NMDS) ordinations performed for each (see Supplemental R Script #1). Only factors for which $p \leq 0.01$ were considered further in analyses, and where a choice between the overlapping PC_URB and Index$_{Combined}$ factors was required, the recommendation of *Hahs & McDonnell (2006)* for Index$_{Combined}$ was adopted.

Boxplots of species richness of the two urban tolerant assemblages were made, binned by an index of urbanisation intensity (Index$_{Combined}$—hereafter urbanisation index) and cover of vegetation (greenspace) (see Supplemental R Script #2).

Species richness of exploiter and adapter species was enumerated for each of 390 grid cells (*Conole & Kirkpatrick, 2011*), along with the index of urbanisation intensity and cover of vegetation. Data were then modelled as hierarchical Bayesian models using R-package 'bayespref' (*Fordyce et al., 2011*) to test the preferences of exploiters and adapters for partitioned urban habitats. Model parameters were estimated using a Markov Chain Monte Carlo (MCMC) approach, with 10,000 MCMC steps following a burn-in of 1,000 generations. The parameters estimated in this way are intended to directly address the hypothesis (*Fordyce et al., 2011*), namely that adapter and exploiter bird assemblages show preferences for urban habitat characterised by differing levels of urbanisation intensity or vegetation cover. The hierarchical Bayesian approach has the advantage of directly estimating the parameter of interest (in this case preference for levels of urbanisation or green space by urban tolerant bird assemblages), and models the uncertainty around those parameters as well as allowing comparisons between *a priori* identified groups, in contrast to methods such as ANOVA or *t*-tests, which assess whether the mean difference is different from zero (*Fordyce et al., 2011*). The estimates are population-level preferences (*Fordyce et al., 2011*).

Within 'bayespref' a facility for assessing model convergence (indicated by MCMChain mixing) by plotting MCMC steps against population level preferences is available (*Fordyce et al., 2011*). A well-mixed chain is one characterised by a broad scatter of data points in the scatterplot without obvious clumping (Figure SR4 in Supplemental R Script #3), whereas clumping of data points indicates poorly-mixed chains. Although a subjective

visual measure, it is sufficient to identify satisfactory MCMChain mixing, and this method was used here to determine when satisfactory model convergence had been achieved.

Proposal distance in the MCMC is set by the 'bayespref' switch 'dirvar'; usually at the default setting of 2. Runs of 'bayespref' with a 'dirvar' value of 2, 5, 10 and 20 were executed, to determine whether optimal mixing of the MCMChains influenced the overall trends in habitat preference (see Supplemental R Script #3), but the gross trends were unchanged. Nonetheless, results cited in this paper use the highest tested proposal distance ('dirvar' = 20) to ensure thoroughly mixed MCMC chains.

Outputs from the 'bayespref' analysis were plotted, with base R functions, as binned median preference with 95% confidence intervals (see Supplemental R Script #4).

Adapter and exploiter species' habitats-of-origin were determined by reference to the literature (*Marchant & Higgins, 1993*; *Higgins & Davies, 1996*; *Higgins, 1999*; *Schodde & Mason, 1999*; *Higgins, Peter & Steele, 2001*; *Higgins & Peter, 2002*; *Higgins, Peter & Cowling, 2006*), and shown in Table 1. Habitat-of-origin is used here to mean the primary natural (pre-urbanisation) habitats that species are known to have occupied. The data for cluster analysis consisted of a standard array, with species as rows and habitat-of-origin as columns (forest, woodland, heath, scrub, urban, farm, air). A Bray-Curtis distance matrix was prepared, and groups of species were formed by hierarchical agglomerative clustering using Ward's algorithm performed on the distance matrix, using core R-function 'hclust' (*R Core Team, 2013*) (see Supplemental R Script #5).

## RESULTS AND DISCUSSION

### Results

In an earlier ordination of all bird species from the Melbourne study, urban exploiters and adapters are shown as overlapping but distinct clusters in ordination space (Fig. S2) (*Conole & Kirkpatrick, 2011*). When the exploiters and adapters were partitioned from the avoiders and run as separate ordinations, different pictures of response to urban environmental factors became apparent (Figs. 1 and 2).

For exploiters the observed species richness vector ($S_{obs}$) was orthogonal with both greenspace and the urbanisation index (Fig. 2). The equivalent vector for adapters (Fig. 1) was orthogonal with the urbanisation index, but almost aligned with that for greenspace (Fig. 2). Greenspace and the urbanisation index were chosen as representative of structural and demographic aspects of urbanisation intensity even though other parameters were included in the initial analyses, and further analyses were limited to these two factors.

The same data plotted as binned boxplots showed that adapter species richness was positively associated with increasing greenspace, but exploiter species richness was flat across the range (Fig. 3). Whilst broadly similar trends were evident for both groups as binned boxplots plotted against the urbanisation index (Fig. 4), adapters trended to zero species richness at the highest levels, whilst 10–15 species of exploiters persisted at the same level. Peak species diversity of urban adapter birds occurred in the middle of the range of urbanisation intensity (Fig. 4). Adapter richness peaked at approximately 0.8 frequency green-space; exploiters at around 0.55 (Fig. 3).

**Table 1** List of bird species analysed in this study.

| Common name | Scientific name | Family | Urban adapter | Urban exploiter | Habitat-of-origin |
|---|---|---|---|---|---|
| White-browed Scrubwren | *Sericornis frontalis* | Acanthizidae | Y | | Forest, woodland, heath, scrub |
| Brown Thornbill | *Acanthiza pusilla* | Acanthizidae | Y | | Forest, woodland, heath, scrub |
| Yellow-tailed Black-Cockatoo | *Calyptorhynchus funereus* | Cacatuidae | Y | | Forest, woodland, heath |
| Gang-gang Cockatoo | *Callocephalon fimbriatum* | Cacatuidae | Y | | Forest, woodland |
| Sulphur-crested Cockatoo | *Cacatua galerita* | Cacatuidae | Y | | Forest, woodland |
| Black-faced Cuckoo-shrike | *Coracina novaehollandiae* | Campephagidae | Y | | Forest, woodland |
| Common Bronzewing | *Phaps chalcoptera* | Columbidae | Y | | Forest, woodland, scrub |
| Australian Raven | *Corvus coronoides* | Corvidae | Y | | Forest, woodland |
| Grey Butcherbird | *Cracticus torquatus* | Artamidae | Y | | Forest, woodland |
| Pied Currawong | *Strepera graculina* | Artamidae | Y | | Forest, woodland |
| Grey Currawong | *Strepera versicolor* | Artamidae | Y | | Forest, woodland, heath |
| Laughing Kookaburra | *Dacelo novaeguineae* | Halcyonidae | Y | | Forest, woodland |
| Rainbow Lorikeet | *Trichoglossus haematodus* | Loriidae | Y | | Forest, woodland, heath |
| Superb Fairy-wren | *Malurus cyaneus* | Maluridae | Y | | Forest, woodland, heath, scrub |
| Eastern Spinebill | *Acanthorhynchus tenuirostris* | Meliphagidae | Y | | Forest, woodland, heath, scrub |
| Bell Miner | *Manorina melanophrys* | Meliphagidae | Y | | Forest, woodland, scrub |
| Noisy Miner | *Manorina melanocephala* | Meliphagidae | Y | | Forest, woodland |
| Spotted Pardalote | *Pardalotus punctatus* | Pardalotidae | Y | | Forest, woodland |
| Tawny Frogmouth | *Podargus strigoides* | Podargidae | Y | | Forest, woodland |
| Crimson Rosella | *Platycercus elegans* | Psittacidae | Y | | Forest, woodland |
| Eastern Rosella | *Platycercus eximius* | Psittacidae | Y | | Forest, woodland |
| Grey Fantail | *Rhipidura albiscapa* | Rhipiduridae | Y | | Forest, woodland |
| Silvereye | *Zosterops lateralis* | Timaliidae | Y | | Forest, woodland, heath, scrub |
| Brown Goshawk | *Accipiter fasciatus* | Accipitridae | | Y | Forest, woodland |
| Galah | *Eolophus roseicapillus* | Cacatuidae | | Y | Woodland, grassland |
| *Rock Dove | *Columba livia* | Columbidae | | Y | Grassland |
| *Spotted Dove | *Streptopelia chinensis* | Columbidae | | Y | Forest, woodland |
| Crested Pigeon | *Ocyphaps lophotes* | Columbidae | | Y | Woodland, grassland |
| Little Raven | *Corvus mellori* | Corvidae | | Y | Woodland, grassland |
| Australian Magpie | *Cracticus tibicen* | Artamidae | | Y | Woodland, grassland |
| Australian Hobby | *Falco longipennis* | Falconidae | | Y | Forest, woodland, heath, scrub |
| Welcome Swallow | *Hirundo neoxena* | Hirundinidae | | Y | Aerial |
| Musk Lorikeet | *Glossopsitta concinna* | Loriidae | | Y | Forest, woodland |
| Little Lorikeet | *Glossopsitta pusilla* | Loriidae | | Y | Forest, woodland |
| White-plumed Honeyeater | *Lichenostomus penicillatus* | Meliphagidae | | Y | Forest, woodland |
| Little Wattlebird | *Anthochaera chrysoptera* | Meliphagidae | | Y | Forest, woodland, heath, scrub |
| Red Wattlebird | *Anthochaera carunculata* | Meliphagidae | | Y | Forest, woodland, heath, scrub |
| Magpie-lark | *Grallina cyanoleuca* | Monarchidae | | Y | Woodland, grassland |
| *House Sparrow | *Passer domesticus* | Passeridae | | Y | Urban, farm |
| *Eurasian Tree Sparrow | *Passer montanus* | Passeridae | | Y | Urban |
| Red-rumped Parrot | *Psephotus haematonotus* | Psittacidae | | Y | Woodland, grassland |
| Willie Wagtail | *Rhipidura leucophrys* | Rhipiduridae | | Y | Woodland, grassland |

*(continued on next page)*

| Common name | Scientific name | Family | Urban adapter | Urban exploiter | Habitat-of-origin |
|---|---|---|---|---|---|
| *Common Starling | *Sturnus vulgaris* | Sturnidae | Y | | Urban, farm, woodland, heath, scrub |
| *Common Myna | *Sturnus tristis* | Sturnidae | Y | | Urban, farm, woodland |
| *Common Blackbird | *Turdus merula* | Turdidae | Y | | Forest, woodland, heath, scrub, urban |
| *Song Thrush | *Turdus philomelos* | Turdidae | Y | | Urban |

**Notes.**

Habitat data from *Marchant & Higgins (1993)*, *Higgins & Davies (1996)*, *Higgins (1999)*, *Schodde & Mason (1999)*, *Higgins, Peter & Steele (2001)*, *Higgins & Peter (2002)* and *Higgins, Peter & Cowling (2006)*.

* Feral species are denoted with an asterisk.

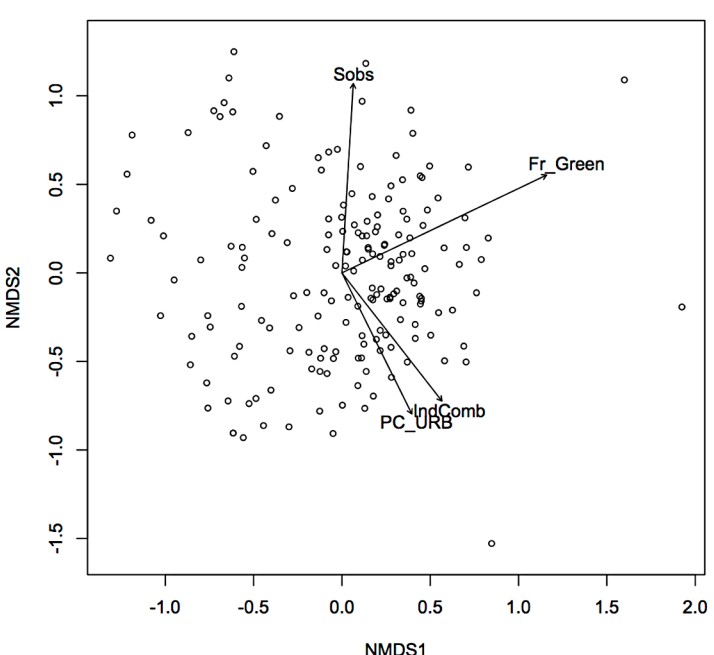

**Figure 1** **Non-metric multidimensional scaling (NMDS) ordination; urban adapters—fitted vectors for which $p \leq 0.01$.**

The hierarchical Bayesian models for greenspace showed a relatively flat preference by urban exploiters across the range; though increasing preference by urban adapters for higher levels of greenspace (median = 0.46; credible intervals 0.424–0.494) almost match exploiter preference (0.54; 0.506–0.576) in the highest bin (Fig. 5; Table S1). Even allowing for high levels of variance in the lower bins where data points were more scarce, the preferences of urban exploiters and adapters did not overlap in any of the greenspace bins.

Hierarchical Bayesian models for the combined index showed a joint preference by urban adapters and exploiters in the middle of the range of the urbanisation index (20.0–29.9). Areas of low (0–19.9) and high (30.0–50.0) urbanisation index were strongly preferred by urban exploiters but not adapters (Fig. 6; Table S2).

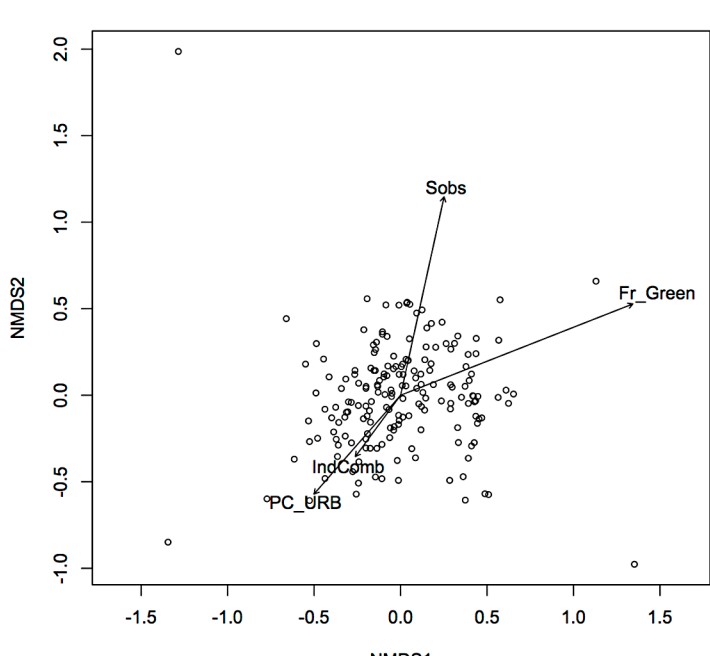

**Figure 2 Non-metric multidimensional scaling (NMDS) ordination; urban exploiters—fitted vectors for which $p \leq 0.01$.**

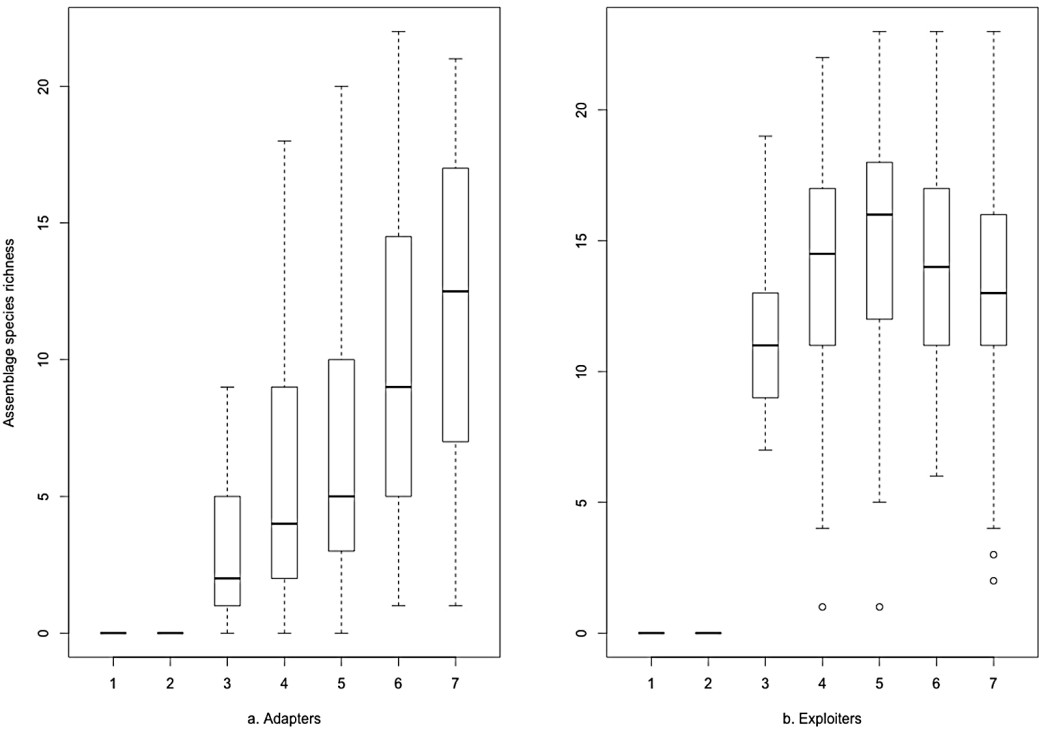

**Figure 3 Species richness of (A) urban adapter and (B) urban exploiter bird species binned by the proportion of Frequency Greenspace at urbanised sites.**

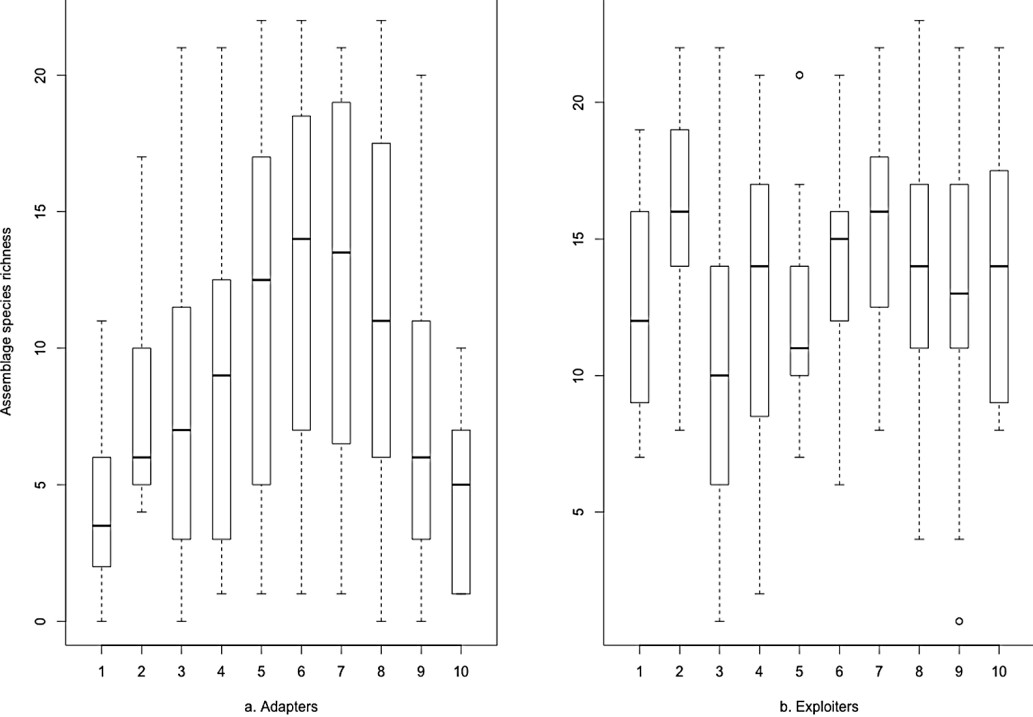

**Figure 4** Species richness of (A) urban adapter and (B) urban exploiter bird species binned by urbanisation intensity (Index$_{Combined}$) at urbanised sites.

The cluster analysis of adapters and exploiters by habitat of origin returned a dendrogram showing two clear major clusters. All of the adapters clustered together in a woody vegetation habitat group, along with a group of exploiters; five indigenous nectarivores (Red Wattlebird *Anthochaera carunculata* (Shaw, 1790), Little Wattlebird *A. chrysoptera* (Latham, 1802), White-plumed Honeyeater *Lichenostomus penicillatus* (Gould, 1837), Musk Lorikeet *Glossopsitta concinna* (Shaw, 1791), Little Lorikeet *G. pusilla* (Shaw, 1970)), two indigenous avivorous raptors (Australian Hobby *Falco longipennis* Swainson, 1837, Brown Goshawk *Accipiter fasciatus* (Vigors and Horsfield, 1827)) and two exotic species which are not exclusively synanthropic (Common Blackbird *Turdus merula*, Linnaeus, 1758, Common Starling *Sturnus vulgaris* Linnaeus, 1758) (*Conole, 2011*). The cluster of exclusively exploiter species were characterised by those originating from open grassy or urban habitats.

The boxplots (Figs. 3 and 4) and the hierarchical Bayesian models (Figs. 5 and 6) showed clear but distinct trends of urban habitat preference by urban exploiter and adapter bird assemblages against these two representative urban habitat measures. The landscape scale preferences of urban adapters and urban exploiters for levels of greenspace never overlap, though they come close to each other at the highest values as exploiter preference declines and adapter preference increases. In contrast, landscape preferences for urbanisation intensity measured by the urbanisation index overlap strongly in the middle of the range but are strongly divergent at the lowest and highest values.

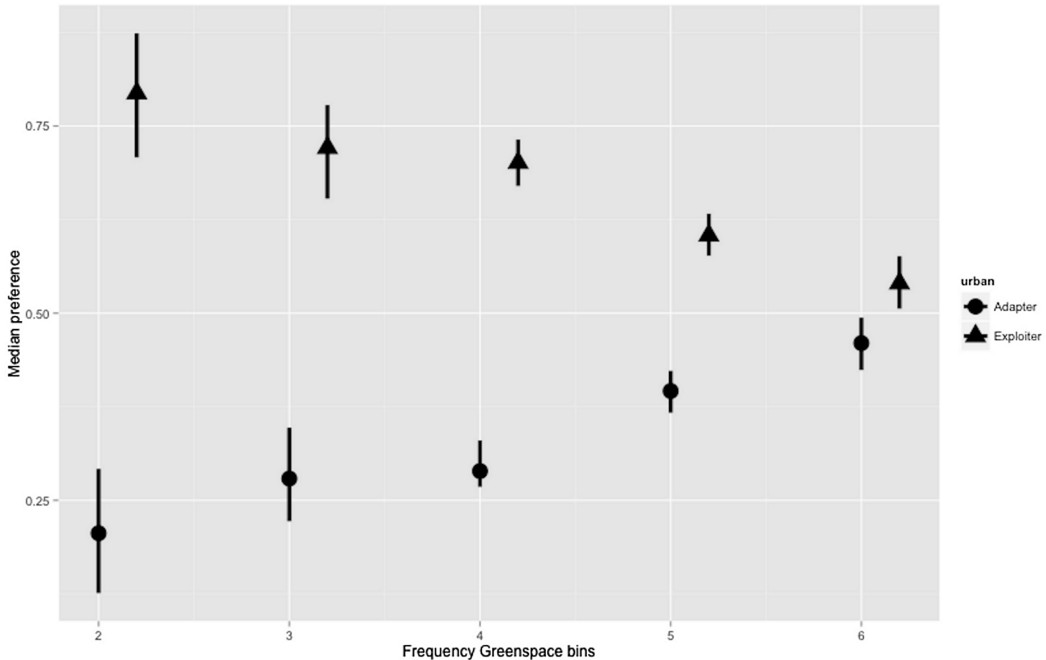

**Figure 5** Posterior density for landscape-scale preferences of urban adapter and exploiter bird assemblages (median preference and 95% credible intervals) binned by Frequency Greenspace at urbanised sites.

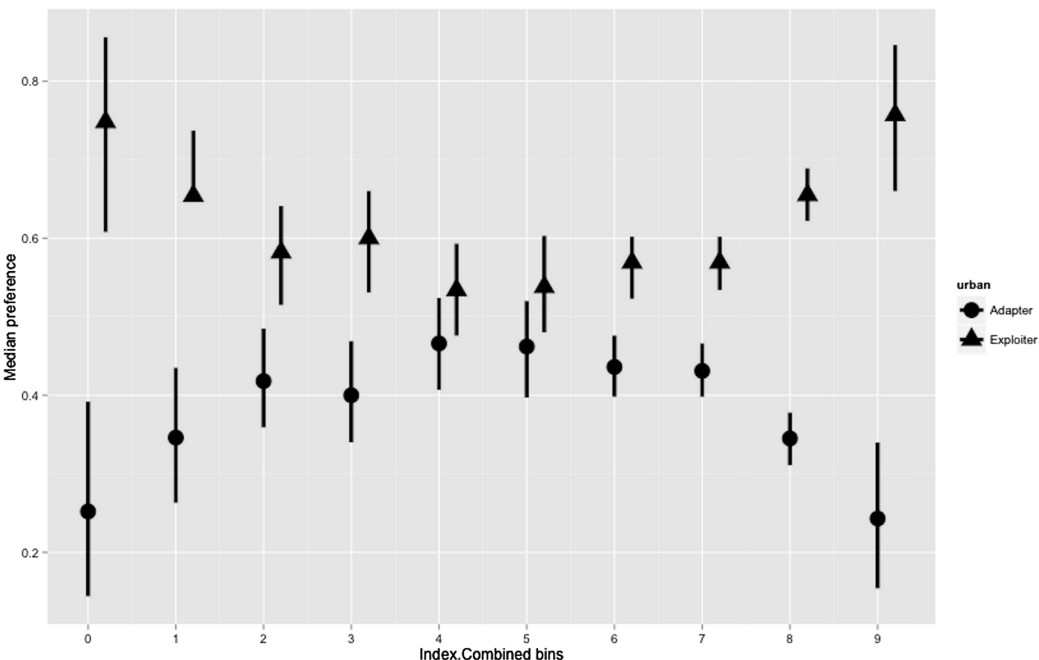

**Figure 6** Posterior density for landscape-scale preferences of urban adapter and exploiter bird assemblages (median preference and 95% credible intervals) binned by urbanisation intensity at urbanised sites.

**Peer**J ___________________________________________________

## Discussion

The diversity of urban adapters on the gradient of urban intensity follows a humped distribution (Fig. 4); the trend even more strongly humped when viewed as landscape scale preference (Fig. 6). This is consistent with the trend seen for urban tolerant birds in other studies (*Marzluff & Rodewald, 2008*; *Tratalos et al., 2007*), and for bird species richness in response to several environmental factors at a landscape scale (*Bar-Massada et al., 2012*). The inverted, humped curve for exploiters is not consistent with the trends for urban tolerant bird species richness seen in other studies (*Tratalos et al., 2007*; *Luck & Smallbone, 2010*; *Shanahan et al., 2014*), and this marks a strong divergence in response by exploiters and adapters to urbanisation intensity.

This quadratic trend in diversity also resembles that described by the Intermediate Disturbance Hypothesis (IDH), where diversity peaks at a midpoint along a gradient of disturbance (*Catford et al., 2012*; *Fox, 2013*). The urban–rural gradient is, however, not a true analogue of a disturbance gradient. Suburban areas are more stable habitats than either the developing fringe or the intensely re-shaped core of the city, and so disturbance itself shows a quadratic distribution along the urban–rural gradient. Also implicit within IDH is a notion of competition/colonisation trade-off amongst species more or less adapted to disturbed environments, and at least for urban adapted birds it has been suggested that competition is not important (*Mikami & Nagata, 2013*) except for specific cases such as the 'despotic' Noisy Miner (*Manorina melanocephala*) (*Kath, Maron & Dunn, 2009*; *Maron et al., 2013*; *Robertson et al., 2013*).

The zone of overlap in habitat preference along the human demographic gradient accords broadly with the inner ring of suburbs in Melbourne; long established and heavily vegetated (*Hahs & McDonnell, 2006*). At the extremes of this gradient lie the new suburbs/exurbia at the fringe, and the central business districts ('down town') at various central locations—either lightly vegetated or with largely treeless vegetation (lawns and pasture) (*Hahs & McDonnell, 2006*). The overlap represents depressed preference by exploiters coincident with greatest preference shown by adapters.

The response of urban tolerant birds to increasing Frequency Greenspace is consistent with wider trends in other cities (*Chace & Walsh, 2006*), and closely mirrors the relationship observed between bird species richness and foliage height diversity observed in a non-urban landscape (*Bar-Massada & Wood, 2014*). Increasing foliage height diversity is a marker of established suburbs versus the developing fringe in Melbourne (*Hahs & McDonnell, 2006*). The distinct responses between adapters and exploiters is also less marked with respect to greenspace than urbanisation intensity.

The responses of the two assemblages to two simple measures of urban habitat character were divergent, consistent with the study's main hypothesis. Though the larger group of urban tolerant bird species may occasionally be treated as one entity, it is clear from this study and others (*Croci, Butet & Clergeau, 2008*; *Catterall, 2009*; *Conole, 2011*; *Conole & Kirkpatrick, 2011*) that the two groups within it are sufficiently distinct in their responses to urbanisation to caution against using pooled data for urban tolerant species in future studies.
The response of urban adapter species to the urbanisation index is consistent with what we broadly understand them to be; adapted to suburbanisation (*Blair & Johnson, 2008*). Greenspace typically increases in old suburbs versus the exurban fringe or downtown areas (*Hahs & McDonnell, 2006*). The strong depression in exploiter preference for mid-range urbanisation intensity (versus the extremes) is less expected. At least with the Melbourne data, there is not a single generalised urban tolerant group of birds. The adapters and exploiters share ecological traits with each other but also with avoiders (*Conole & Kirkpatrick, 2011*).

In part the contemporary avifauna of an urbanised area is a legacy of the species present in the former landscape, rather than solely being the product of invasion or colonisation (*sensu Møller et al., 2012*). As urban areas progressively come to resemble woodland, structurally if not floristically (*Kirkpatrick, Daniels & Zagorski, 2007*), it makes sense that the urban tolerant bird species are likely to include legacy woodland-adapted species. Despite the findings of *Blair & Johnson (2008)* in North American urban areas, it does not appear that suburban areas within a previously forested landscape in Melbourne are loci for indigenous woodland bird extirpation or exotic bird invasion (*Conole & Kirkpatrick, 2011*). Instead the reverse seems to be true. They are sites for colonisation and expansion of some indigenous woodland birds (adapters) and places where exotic exploiters are less abundant.

Exploiters are mostly indigenous species derived from open environments such as grassland and grassy open-woodland (*Møller et al., 2012*), with a small cohort of synanthropic exotic species and indigenous dietary specialists (avivorous raptors, nectarivores) more typical of forest/woodland habitats (Table 1; Fig. 7) (*Conole & Kirkpatrick, 2011*). Adapters as a group are all indigenous species of forest, woodland and riparian scrub origins (Table 1; Fig. 7), and they have closer affinities with the riparian and bush remnant urban avoiders than the exploiters (*Conole & Kirkpatrick, 2011*). It is therefore remnants of the former indigenous avifauna of wooded parts of Melbourne that are the source of the emerging group of urban adapted species, though none are yet as successful as the aptly named urban exploiters. The adapters are essentially the vanguard of a group of semi-specialised bird species that utilise particular niches of greater foliage height diversity within urban matrix habitats, but are not yet ubiquitous across the matrix in the way of exploiters.

The responses observed here of each group to both degree of urbanisation and greenspace are largely explained by their ecological histories. The exploiters are able to use disturbed habitats across the matrix analogous to their original habitats, and many of them were established in Melbourne during the early stages of urban expansion and consolidation of the city. As suburban parts of the city became more heavily vegetated and less open, a group of species from analogous riparian/forest habitats became increasingly well established in parts of the city proximate to their source natural habitats. Many parts of the urban matrix are now at or close to the point of saturation with members of the exploiter assemblage due to their ubiquity, but the number of adapter species contributing to bird species richness at points across the matrix is likely to increase on a site by site basis as the process of afforestation of the older suburbs continues. It follows then that the

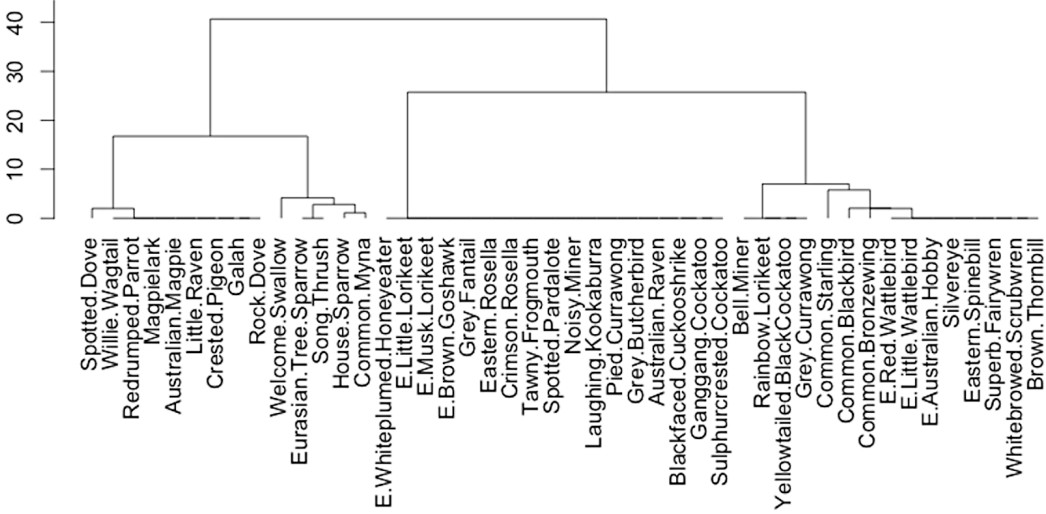

**Figure 7** Cluster dendrogram (Ward method) of adapters and exploiters by habitat-of-origin. Exploiters that cluster within the adapters are prefixed with the letter "E".

distribution of exploiter species may decline in more established suburban parts of the city over time, though expanding in range and continuing to dominate in developing areas of the city at or near the fringe. Clues to this trend can be found in studies that model the trajectory of abundance for open habitat, ground feeding specialists (such as the Crested Pigeon *Ocyphaps lophotes* GR Gray, 1842) declining as those habitats become denser with woody vegetation (*Kutt & Martin, 2010*).

## CONCLUSION

The partitioning of adapters and exploiters within the urban tolerant grouping in this study reveals the possible pitfall in assuming uniformity of response of all 'urban tolerant' species, that otherwise might result in the overlooking of a key to understanding how habitat origins may be important for understanding bird species' adaptation to urban environments. Other workers have examined the importance of a variable suite of physiological and behavioural traits that may predispose birds to urban adaptability (e.g., *Kark et al., 2007*; *Møller, 2009*; *Evans et al., 2010*). This study has examined the higher order habitat filtering mechanism that may be influential in this regard, and more broadly generalisable as a conceptual model at the scale of the landscape and the assemblage.

In a similar way to that in which time since establishment has been found to be related to high urban densities of some bird species (*Møller et al., 2012*), or biogeographic origin predictive of urban adaptation extent (*González-Oreja, 2011*), spatial and habitat origins of members of bird assemblages influence the degree to which they become urban tolerant; ranging from not at all through to ubiquitous. Bird species that classify as urban tolerant will further classify as either exploiters or adapters according to the degree of openness of their habitats-of-origin.

## ACKNOWLEDGEMENTS

This study forms part of a larger doctoral research project on Melbourne's urban avifauna by the author at the University of Tasmania. Professor Jamie Kirkpatrick (Geography & Environmental Studies, University of Tasmania, Australia) provided insights and constructive criticism along the way. Andrew Silcocks and Dr Mike Weston facilitated access to the BirdLife Australia 'Atlas II' database. Dr Amy Hahs (ARCUE, University of Melbourne, Australia) provided access to her dataset of remotely sensed landscape metrics. Dr Kath Handasyde (Department of Zoology, University of Melbourne) provided support during the writing of this paper.

### Funding

This work is self-funded.

### Competing Interests

I am not aware of any competing interests that are relevant for this work.

### Author Contributions

- Lawrence E. Conole conceived and designed the experiments, performed the experiments, analyzed the data, contributed reagents/materials/analysis tools, wrote the paper, prepared figures and/or tables, reviewed drafts of the paper.

### Supplemental Information

Supplemental information for this article can be found online at http://dx.doi.org/10.7717/peerj.306.

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
