# Peer review of "Degree of adaptive response in urban tolerant birds shows influence of habitat-of-origin"

_PeerJ, doi:10.7717/peerj.306_

## Round 0.1 · original submission · Major Revisions

Dear Lawrie,

i am inviting you to resubmit a major revision of your paper which will be subject to additional peer review by new reviewers. I believe the work you have undertaken to merit publication but share the concerns of the reviewer with respect to the clarity of the manuscripts. The revision you have provided by email is a significant improvement but more work is still required to address the concerns of the reviewer and myself.

Could i ask you to thoroughly cross check methods, figures, results and supplementary material to ensure that code, figures and results are presented sequentially as laid out in the methods section. The code included in your new supplementary material is a very useful addition, but you only need to display the heads of your data frames, and additional code for your hierarchical models and the cluster analysis would be useful.

I would also suggest that you shorten the introduction and relate it more explicitly to your hypotheses. The first hypothesis could also be expressed in language that general biologists would be more able to interpret.

Add an explanation of the assessment of model convergence (Gelman Rubin diagnostics) or discuss the lack of such assessment.

The figures need keys and should be cited in numerical order in the text.

The selection procedure for variables in the ordination needs to be more explicit. Summarise the results of ordination with all variables (with reference to the figures in your new supplementary material) and justify the selection of variables.

Acknowledge uncertainty in response (large variance) to greenspace for both groups

The discussion is great, but maybe tone down the language with respect to the conclusion. Fallacy is a bit strong in my view...

A conceptual figure could also help explain how you view urban tolerant species and why it matters.

I appreciate that we are asking for a lot of work (and that further reviewers may make further suggestions) and am aware that as an author this can be frustrating. However, I would strongly encourage you to persist with this manuscript in order to realise its potential merit. I would like to see it published in peerJ.

Reviewer 1 ·

Basic reporting

Although the study is interesting, I will not recommend this manuscript to be published. This script still needs a lot of work. I have not finished correcting because I think that work is a draft. It has basic errors in label and caption of figures, legends, scientific names, acronyms, interpretation of results, species' table, etc. Additionally, the work seems to be part of another publication, not an independent publication.

Experimental design

No Comments

Validity of the findings

No Comments

Additional comments

No Comments

---

## Round 0.2 · accepted · Accept

Thank you for constructively engaging with the peer review process and amending your manuscript in line with the suggestions of the reviewers and my editorial comments. The rebuttal of the remaining points is appropriate and i'm happy to accept your article for publication